# The Neurotoxic Effect of Ochratoxin-A on the Hippocampal Neurogenic Niche of Adult Mouse Brain

**DOI:** 10.3390/toxins14090624

**Published:** 2022-09-06

**Authors:** Eva Mateo, Rik Paulus Bernardus Tonino, Antolin Canto, Antonio Monroy Noyola, Maria Miranda, Jose Miguel Soria, María Angeles Garcia Esparza

**Affiliations:** 1Department of Microbiology and Ecology, School of Medicine and Dentistry, University of Valencia, 46001 Valencia, Spain; 2Haematology Department, Leiden University, 1043 AJ Leiden, The Netherlands; 3Department of Biomedical Sciences, Cardenal Herrera University-CEU Universities, 46001 Valencia, Spain; 4Neuroprotection Laboratory, Faculty of Pharmacy, Autonomous University of the State of Morelos, Cuernavaca, Morelos 98100, Mexico; 5Department of Pharmacy, Cardenal Herrera University-CEU Universities, 46001 Valencia, Spain

**Keywords:** ochratoxin A, brain, hippocampus, neurogenic niche, neurotoxicity, cell morphology

## Abstract

Ochratoxin A (OTA) is a common secondary metabolite of *Aspergillus ochraceus*, *A. carbonarius*, and *Penicillium verrucosum*. This mycotoxin is largely present as a contaminant in several cereal crops and human foodstuffs, including grapes, corn, nuts, and figs, among others. Preclinical studies have reported the involvement of OTA in metabolic, physiologic, and immunologic disturbances as well as in carcinogenesis. More recently, it has also been suggested that OTA may impair hippocampal neurogenesis in vivo and that this might be associated with learning and memory deficits. Furthermore, aside from its widely proven toxicity in tissues other than the brain, there is reason to believe that OTA contributes to neurodegenerative disorders. Thus, in this present in vivo study, we investigated this possibility by intraperitoneally (i.p.) administering 3.5 mg OTA/kg body weight to adult male mice to assess whether chronic exposure to this mycotoxin negatively affects cell viability in the dentate gyrus of the hippocampus. Immunohistochemistry assays showed that doses of 3.5 mg/kg caused a significant and dose-dependent reduction in repetitive cell division and branching (from 12% to 62%). Moreover, the number of countable astrocytes (*p* < 0.001), young neurons (*p* < 0.001), and mature neurons (*p* < 0.001) negatively correlated with the number of i.p. OTA injections administered (one, two, three, or six repeated doses). Our results show that OTA induced adverse effects in the hippocampus cells of adult mice brain tissue when administered in cumulative doses.

## 1. Introduction

The mycotoxin ochratoxin A (OTA), or 7-carboxy 5-chloro-8-hydroxy-3, 4 dihydro-3-R-methylcoumarin-7-L-β-phenylalanine (Figure 1), is a common metabolite produced by *Aspergillus ochraceus*, *A. carbonarius*, and *Penicillium verrucosum* [1].

OTA is rapidly absorbed and distributed but slowly eliminated and excreted, leading to potential accumulation in the body, which is due mainly to its binding to plasma proteins and a low metabolism rate. Plasma half-lives range from several days in rodents and pigs to several weeks in nonhuman primates and humans [2,3].

OTA has been found in barley, oats, rye, wheat, cereals, grains, beans, corn, spices, and products such as coffee, grape juice, wine, beer, and bread [4,5]. In vitro and in vivo studies have shown that this toxin induces immune toxicity, hepatotoxicity, nephrotoxicity, and reproductive and developmental toxicity [6].

Thus, previous studies have described that severe health hazards are associated with mycotoxin exposure, their molecular signaling pathways and processes of toxicity, and their genotoxic and cytotoxic effects on humans and animals [7]. For humans, however, hazard identification has been more difficult. Several adverse human health effects, including the kidney diseases Balkan endemic nephropathy (BEN) and chronic interstitial nephropathy (CIN), have been associated with exposure to OTA; however, these associations have thus far been less conclusive than those for OTA-associated adverse effects in laboratory animal studies [8].

Furthermore, because it causes damage at the molecular level (including causing single-strand DNA breaks, covalent DNA adduction, and DNA oxidation), it has been reported as a possible xenobiotic carcinogen in humans [9,10,11] and as a teratogenic agent in several laboratory and farm animals, including rats, mice, hamsters, quails, and chickens [12].

Several experimental studies have suggested that OTA also has a neurotoxic effect in humans. Indeed, OTA, as well as its non-toxic corresponding metabolite, ochratoxin α (OTα) [13], has been identified and quantified in human blood and urine [14,15,16,17]. In turn, studies in rat and porcine brain capillary endothelial cells have shown its permeability, allowing it to reach the brain [17,18]. However, the cellular mechanisms responsible for the neurotoxicity of OTA have not been clearly elucidated, although oxidative stress, DNA damage, and mitochondrial dysfunction are plausible possibilities [19,20,21,22]. These toxic mechanisms are implicated in neurodegenerative disorders such as Alzheimer’s and Parkinson’s disease, and thus, OTA is considered an environmental factor that could contribute to the development of oxidative stress and neurodegeneration [21].

Sava et al. [23] suggested that OTA impairs hippocampal neurogenesis in vivo and that this might be associated with learning [24] and memory deficits [23,25]. The neurotoxicity of OTA has been shown to be most pronounced in the ventral mesencephalon, hippocampus, and striatum brain regions [26], even when its bioconcentration was significantly lower than in other regions of the brain. In addition, the half-life of OTA in the hippocampus was found to be only 42.5 h [27]. In turn, the hippocampus (a primary site of neurodegeneration in Alzheimer’s disease) exhibited relatively low OTA levels with concurrently pronounced OTA neurotoxicity [26]. In this context, it has been hypothesized that low-level exposure to OTA may exert delayed neurotoxic effects which could, in turn, contribute to the development of neurodegenerative disorders.

After the tolerable daily intake dose of OTA was set to 3 ng/kg of body weight, the EU Scientific Committee on Food recommended a reduction in human exposure as low as reasonably possible due to concerns regarding the potential genotoxicity and carcinogenicity of this mycotoxin [1,3]. Extensive data have been published on the toxic effects of OTA, but very little has been published regarding long-term in vivo hippocampal exposure to this mycotoxin. In this context, it would be useful to elucidate the neurotoxicity effects in long-term research. Short-term in vitro studies with high doses have shown that neural stem/progenitor cells are vulnerable to OTA [28], but data on its long-term neurotoxic effect in vivo are still lacking. Such long-term studies might show whether physiological levels of OTA exert neurotoxicity through oxidative stress. Thus, the aim of this study was to perform an 18-day in vivo toxicity assay to investigate the long-term effects of OTA exposure on neurogenesis in the hippocampal dental gyrus of adult mouse brains.

## 2. Results

### 2.1. Ochratoxin A Reduces the Number of Astrocytes in the Dentate Gyrus

Twenty-one animals were treated with increasing accumulative doses of 3.5 mg of OTA (n = 17) or vehicle (n = 4) per kilogram of body weight and were sacrificed 3 days after the last scheduled injection. Immunofluorescence staining for GFAP was performed to determine whether the OTA had influenced the number of astrocytes present in the hippocampus (Figure 2). Astrocytes (type B cells) are precursors to progenitor cells (type C cells), which in turn are precursors of migrating neuroblasts (type A cells). Furthermore, astrocyte activity is crucial for the correct formation and function of the blood−brain barrier, which are vital for providing the appropriate environment for proper neuronal functioning and protecting the central nervous system from injury and disease.

As shown in Figure 2, compared to the controls, we found a significant decrease in GFAP cell expression in the dentate gyrus after treatment with OTA. The correlation between the number of doses and the decrease in GFAP-labeled cells indicated that this effect was dose-dependent. After six cumulative doses, there was a dramatic decrease (by more than 60%) in countable cells compared to the controls, whereas one injection of OTA only decreased the number of countable cells by 11.8% (control = 3996 cells/mm^2^; OTA1 = 3523 cells/mm^2^; OTA6 = 1591 cells/mm^2^). Compared to the controls, there were significantly fewer countable cells in the OTA1 (*p* < 0.01) and OTA2, OTA3, and OTA6 (*p* < 0.001). All groups differed significantly from one another (*p* < 0.01). No significant differences were found between OTA1 and OTA2.

Furthermore, we observed an apparent morphological change in the cell structure, as shown in Figure 3. Astrocytic processes seemed less profound in treated animals, and as the number of doses increased, the astrocytes became difficult to morphologically identify. The statistical significance of this change compared to the control was calculated using one-way ANOVA with a post-hoc LSD test. Figure 4 shows the results obtained in the quantitative morphological astrocyte study. The astrocytes from the control animals were significantly longer and more ramified than those from animals that received three or six OTA doses. The administration of two doses of OTA did not affect the number of branches per cell or the branch lengths. These results agree with those obtained for GFAP expression (Figure 2) and may confirm the possible neurotoxic effects of OTA.

### 2.2. Ochratoxin A Reduces the Number of Young Neurons in the Dentate Gyrus

Once we determined that OTA negatively affected astrocytes, we examined the vulnerability of young neurons (type A cells) to this mycotoxin. Counting the DCX-stained cells in the DG revealed a significant difference in all the groups compared to the control (*p* < 0.001; Table 1 and Figure 5). However, the differences were less clear for astrocytes. After one and six doses of OTA, 16% and 38.7% fewer cells were counted, respectively (control = 3825, OTA1 = 3217, and OTA6 = 234 cells/mm^2^). Again, no significant difference was detected when comparing OTA1 and OTA2. Nonetheless, OTA3 and OTA6 significantly differed from OTA1 and OTA2, while OTA6 had significantly fewer DCX-positive cells (*p* ˂ 0.05) compared to OTA3. In terms of astrocytes, small morphological changes, albeit fewer evident alterations, were observed for the young neurons (Figure 6). In addition, there was also an interesting difference in the fluorescence, with the overall intensity and single-cell level fluorescence being reduced. We calculated the statistical significance compared to the control using one-way ANOVA, applying LSD as the post-hoc test. These data are expressed as the mean ± standard error.

### 2.3. Ochratoxin A Reduces the Number of Mature Neurons in the Dentate Gyrus

Given the finding that OTA reduces young neurons and astrocytes, which are both types of cells that can proliferate, we examined whether mature neurons were also influenced by OTA, with significant differences found in all the groups (*p* < 0.001). Notably, the decrease in the countable mature neurons was more striking than the decrease in young neurons and astrocytes: 31.6% and 62.4% of the labeled cells had perished after 1 and 6 doses of OTA, respectively (control = 7990, OTA1 = 5466, and OTA6 = 3006 cells/mm^2^; Table 2 and Figure 7).

Interestingly, we did not find any statistically significant differences when comparing OTA1 to OTA2 and OTA3 to OTA6. This might indicate that the acute mature neuron response to OTA is more prominent than the chronic effect or that there is a group of neurons that perish at low concentrations of OTA while others are not that vulnerable to this mycotoxin. Another interesting finding was that in the case of MAP2 labeling, apart from the changed morphology (Figure 7), the fluorescence intensity was also lower than for DCX or GFAP labeling, as shown in Figure 8. Whereas in the control the fluorescence intensity was high, in OTA6, even the best-labeled cells were hard to identify. The statistical significance of the fluorescence intensity compared to the control group was calculated using a Games–Howell test (no-homogeneous variance post-hoc test), and the data are expressed as the mean ± standard error. 

## 3. Discussion

The mycotoxin OTA, which is found in many foodstuffs [4,5], produces a toxic effect in various organisms. It has been described as carcinogenic, nephro-, immuno-, and genotoxic, is related to increased oxidative stress [6], and is considered neurotoxic [25,26,27,29,30]. For humans, however, hazard identification has been more difficult. Several adverse human health effects, including the kidney diseases Balkan endemic nephropathy (BEN) and chronic interstitial nephropathy (CIN), have been associated with exposure to OTA; however, these associations have thus far been less conclusive than those for OTA-associated adverse effects in laboratory animal studies [8].

The relationship between OTA and neurotoxicity has been established in vitro [19,31,32]. However, very little has been published about the neurotoxic effects of OTA in in vivo pre-clinical studies.

In this current study, we demonstrated how OTA can have a significant detrimental impact on the hippocampus of adult mice. In vivo OTA significantly affected astrocytes (neuronal stem cells) and young and mature neurons after one dose (3.5 mg/kg body weight). Fewer GFAP/DAPI co-labeled cells were countable per square millimeter, with a decrease of 11.8% after one dose of OTA and up to 61.3% after six doses. Furthermore, DCX/DAPI co-labeled cells decreased by 16% and 38.7% after one and six doses, respectively. In the case of Map2/DAPI co-labeled cells, there were 31.6% and 62.4% less after one and six doses, respectively. Since consecutive doses caused more damage, we now know that chronic exposure to OTA increasingly affects neural precursors as well as differentiated neurons.

It appears that young neurons were less vulnerable than stem cells and mature neurons, given that 61.3% of the young neurons survived the six doses of OTA compared to 37.6% and 39.8% for mature neurons and astrocytes, respectively. Together with the subventricular zone, the subgranular zone (SGZ) of the dentate gyrus serves as a source of neural stem cells in the process of adult hippocampal neurogenesis [33]. Unlike the neurons of the subventricular zone, the newly generated neurons in the SGZ do not migrate out of the dentate gyrus. Hence, the effect of OTA on neurogenesis in the hippocampus can be measured by focusing only on the dentate gyrus. Thus, to the best of our knowledge, this is the first in vivo demonstration that the dentate gyrus was negatively affected by chronic exposure to OTA.

Type A, B, and C cells are involved in neurogeneration. Type B cells (astrocytes) generate type C cells (proliferative precursors), which in turn give rise to type A cells, or migrating neuroblasts [34,35,36]. Thus, any negative effect on type B cells might lead to a decrease in type C cells and, consequently, deprive the type A cell population. Zurich et al. [37] described that OTA affects the neuroprotective capacity of glial cells by inhibiting GFAP with a consequent decrease in GFAP-positive cells, indicating a deprivation of type B and C cells. Because type A cells are derived from type C cells, the reduction of DCX-positive cells found in our study correlates with the findings of Zurich et al. [37].

Given that MAP2-labeled mature neurons were also affected, in our work, an inverse linear correlation between the OTA doses and the labeled cell count was observed for every cell marker we employed. Indeed, in previous in vitro and in vivo research, dose-dependent relationships were also found in other neurogenic zones when using OTA [19,22,25,32]. Based on this knowledge, we decided to look for a similar correlation when cumulative OTA doses were repetitively administered over time; our results show a dose-dependent relationship between the administration of OTA and cell survival.

An interesting observation was that there was no significant difference between the OTA1 and OTA2 groups for any of the three immunolabels. We hypothesized that the accumulated damage might not just be dose-dependent but could also act via a more complex system involving the blood−brain barrier. As reported by Sava et al. [27], i.p. treatment with one dose of 3.5 mg OTA/kg body weight led to significant changes in hippocampal neurogenesis, a result that also correlates with our findings after one dose of OTA. Surprisingly, our results also show that a second dose of OTA did not impact the neurons as much as we expected, probably because of the ability of the brain to recover and the half-life of 42.5 h for OTA in the hippocampus. This suggests that OTA does not accumulate in the dentate gyrus.

Alternatively, the difference in morphology we observed in OTA2 compared to OTA1 could indicate impaired astrocyte function after exposure to OTA. Because astrocytes are important for the function of the blood−brain barrier [38,39], impaired astrocyte function might lead to severe deregulation of its main functions in the support of the blood−brain barrier. These functions include the removal of toxic residues and controlling glycogen accumulation [40], which could possibly cause less efficient expulsion of OTA from the brain.

A reduction in the number and length of astrocyte branches was observed after brain hypoxia/ischemia insults, and 72 h after a hypoxia/ischemia insult in aged rats, the brain astrocytes had fewer and shorter branches than those in control animals [41]. Another work using the same treatment also confirmed a reduction in the number of branches in aged astrocytes [42]. These authors suggested that these astrocyte alterations may be related to the impairment of these cells in providing support to neurons. In turn, this may decrease the strength of the synapses and alter the metabolism of several neurotransmitters.

In this current study, we demonstrated an OTA-related reduction in the length and number of astrocyte branches. Importantly, these adverse effects were only statistically significant after the administration of three or six doses of OTA but not after receiving only two OTA doses. OTA likely has a longer half-life in the hippocampus, and thus, impaired astrocyte function could allow it to accumulate to higher concentrations, thereby increasing its neurotoxicity. Taken together, this could perhaps explain why a second dose of OTA did not lead to any significant differences compared to a single dose, even though altering the underlying processes would enable subsequent doses to cause measurable effects.

In this report, we did not study the selectivity of OTA toxic effects on astrocytes and neurons [32]. However, we did find that the decrease in the number of young neurons caused by OTA was not as severe as for astrocytes or mature neurons, especially after six doses. This could indicate that type B and C cells and mature neurons are more vulnerable to OTA than type A cells. In the longer term, we might expect the number of young neurons to decrease more because they are derived from type C cells, which seem to be more directly affected by OTA. Of course, based on our work, we cannot exclude the possibility that changes in hippocampal volume after OTA treatment might have contributed to the reduction we observed in cell numbers per square millimeter.

Nonetheless, our findings demonstrate an adverse effect of chronic levels of OTA on the cell survival of astrocytes and neurons in the dentate gyrus, leading to a progressive reduction of neurogenic capacity. Whether these changes are long-lasting and how they affect hippocampal function merits further investigation. In this study, we demonstrated that, at a cellular level, the hippocampus can be severely damaged by OTA. In this study, experiments with hematoxylin and eosin were performed, and no significant macro-structural changes were observed; however, new studies should be performed using electron microscopy in order to study the ultrastructure of the hippocampus after OTA exposure.

Thus, on the one hand, the effect of OTA on the function of the hippocampus should be further examined and behavioral studies should be performed. On the other hand, no work has yet been done to study the recovery of the hippocampus after OTA treatment lasting longer than three days. Therefore, it will be important to discover whether the hippocampus can reverse the damage done by OTA in order to understand the potential neurotoxicity of OTA.

## 4. Materials and Methods

### 4.1. Experimental Animals

Twenty-one male C57BL/6 mice (weighing 21 ± 4 g, 5–6 months old) obtained from Harlan (Barcelona, Spain) were used. The animals were housed five per cage under controlled temperature (24 ± 1 °C) and humidity (60 ± 1%) conditions, with a 12 h light/dark cycle and water and food ad libitum. Both hippocampal regions of each mouse were used. All animal procedures were conducted in accordance with Spanish legislation (RD 1201/05) and the guidelines of the CEU Cardenal Herrera University Animal Care Committee, approval number 117021.

### 4.2. Ochratoxin A Administration in the In Vivo Assay

OTA, as shown in Figure 1 (1 mg/mL, stock solution, 99% purity), was purchased from Sigma-Aldrich (St. Louis, MO, USA), dissolved in 100% ethanol, and further diluted to a concentration of 0.1 M (pH 7.2) in phosphate-buffered saline (PBS). Seventeen mice received an i.p. dose of 3.5 mg OTA/kg body weight (equivalent to approximately 10% of the LD50) in a volume of 2.8 µL/g body weight. This dose was selected because it is in the range of doses used by other studies; in this way, the obtained results could be compared easily [30,43]. The animals were divided into four experimental groups with n = 4–5 animals each: OTA1, OTA2, OTA3, and OTA6, which each received one, two, three, or six doses, respectively. Because the hippocampal half-life of OTA is 42.5 h [27], each dose was separated by three days to minimize its cumulative toxic effect. A control group of mice was injected i.p. with the vehicle from one to six times, corresponding with the treated mice. Three days after the last OTA dose administration, all the mice were humanely euthanized by cervical dislocation.

### 4.3. Immunohistochemistry

The animals were transcranial perfused with 4% paraformaldehyde. Their brains were removed and postfixed overnight in the same fixative solution, and then washed and cryoprotected by immersion in 30% sucrose dissolved in PBS for 2 days at 4 °C. Coronal sections (20 µm) were serially obtained using a cryostat (Leica, Wetzlar, Germany), mounted on glass slides, and stored at −20 °C. The brain sections between the Bregma zone (1.46 mm) and the interaural area (2.34 mm) and the Bregma zone (2.32 mm) and the interaural area (1.50 mm) were selected using the Paxinos G atlas [44].

The brain sections were washed in PBS and blocked for 2 h with blocking buffer (BB; 10% fetal bovine serum in PBS-triton X-100 0.1%) at room temperature.

Next, the sections were incubated overnight at 4 °C with either rabbit polyclonal anti-Glial fibrillary acidic protein (GFAP: Dako, Denmark), anti-microtubule-associated protein 2 (MAP2: Millipore, California), or rabbit polyclonal anti-neural migration protein doublecortin primary antibody (DCX: Abcam, Cambridge, UK) diluted in BB (GFAP, 1:500; Map2, 1:200; DCX, 1:300). After rinsing with PBS-triton X-100 0.1%, the binding of the primary antisera was visualized with Alexa Fluor™ 488 goat anti-rabbit and Alexa Fluor™ 594-conjugated donkey anti-rabbit (1:200, both Invitrogen, Barcelona, Spain) diluted in BB. Secondary antibodies were applied for 2 h at room temperature in the dark. Afterwards, the sections were mounted with DAPI Vectashield (Vector Labs, Peterborough, UK) and coverslipped. In the case of Map2 and DCX staining, the tissue sections were incubated at 100 °C in 10 mM citrate (pH 8.0) for 20 min to expose the immunoreactive sites before adding the primary antibody. The antibodies and specifications are shown in Table 3.

### 4.4. Cell Quantification and Statistical Analysis

The cells were counted in the granular and polymorphic layers of the dentate gyrus using digital images of the hippocampus obtained with a DS-Fi-1 (Nikon, Spain) digital camera coupled with a Leica DM2000 microscope. Four animals from each group were analyzed using four serial coronal sections per animal at 100 µm intervals. Only cells stained positively for both DAPI and one other marker were included. The counting area was measured using ImageJ 1.46r software. The data are expressed as the total number of immunostained cells/mm^2^.

To quantify the number and length of the branches, the fluorescence photomicrographs were converted into skeletonized images and analyzed using the ImageJ software plugins AnalyzeSkeleton and FracLac, according to the method described by Young and Morrison [45]. One-way ANOVA was used to analyze the statistical significance of any between- and within-group differences using SPSS statistics software (Version 17.0, SPSS Inc., Chicago, IL, USA). When variances were found to be homogenously distributed, a least significant difference test (Fisher’s LSD) was used. When variances were not homogenously distributed, as was the case with counting the MAP2 cells, a Games–Howell test was used. Any differences were considered significant at *p* < 0.05.

## Figures and Tables

**Figure 1 toxins-14-00624-f001:**
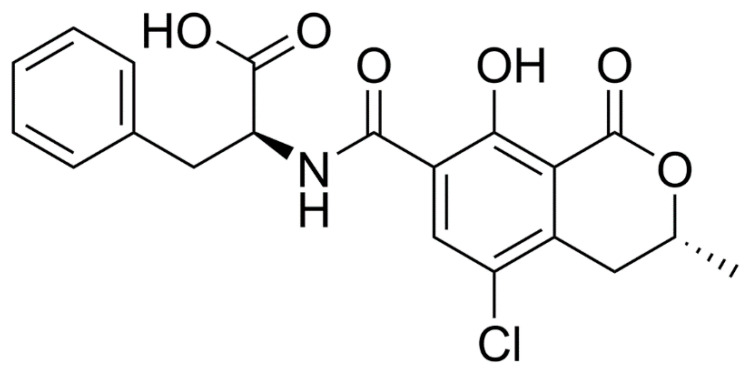
The chemical structure of ochratoxin A.

**Figure 2 toxins-14-00624-f002:**
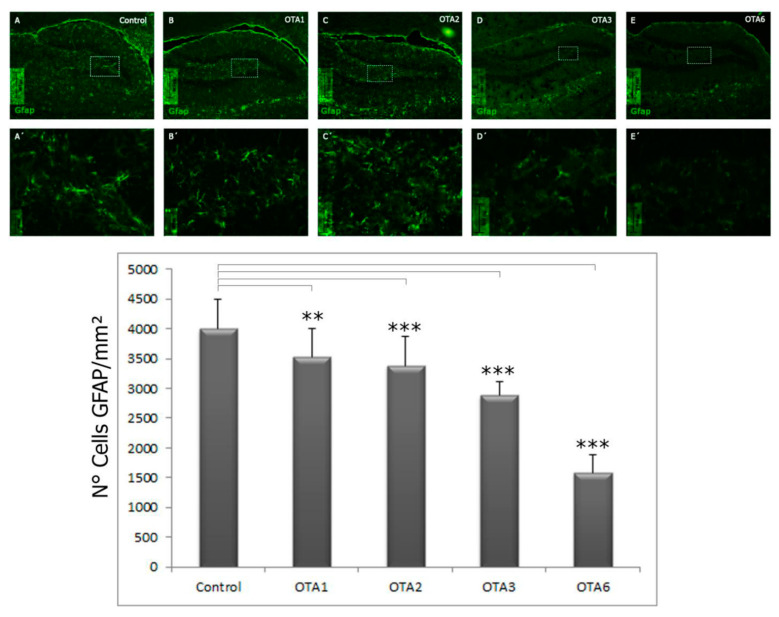
Photomicrographs of GFAP-labeled cells in the control, OTA1, OTA2, OTA3, and OTA6 groups (which received one to six injections of ochratoxin A, respectively), taken at 20× magnification. Compared to the control, the number of GFAP-positive cells decreased as the number of ochratoxin A treatments increased. (Control n = 4, OTA1 n = 4, OTA2 n = 4, OTA3 n = 4, OTA6 n = 5). ** = *p* < 0.01, and *** = *p* < 0.001.

**Figure 3 toxins-14-00624-f003:**
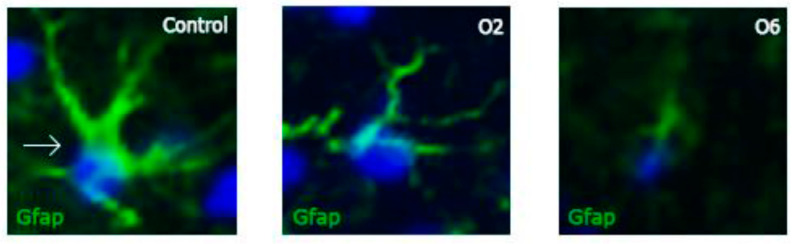
Individual GFAP-labeled cells in the control group and after two (O2) or six (O6) doses of ochratoxin A. Note how the morphology of these cells changed as the number of doses increased, with the cellular body appearing to decrease in volume and the cytoplasmic processes retracting. These photomicrographs were obtained at a 40× magnification.

**Figure 4 toxins-14-00624-f004:**
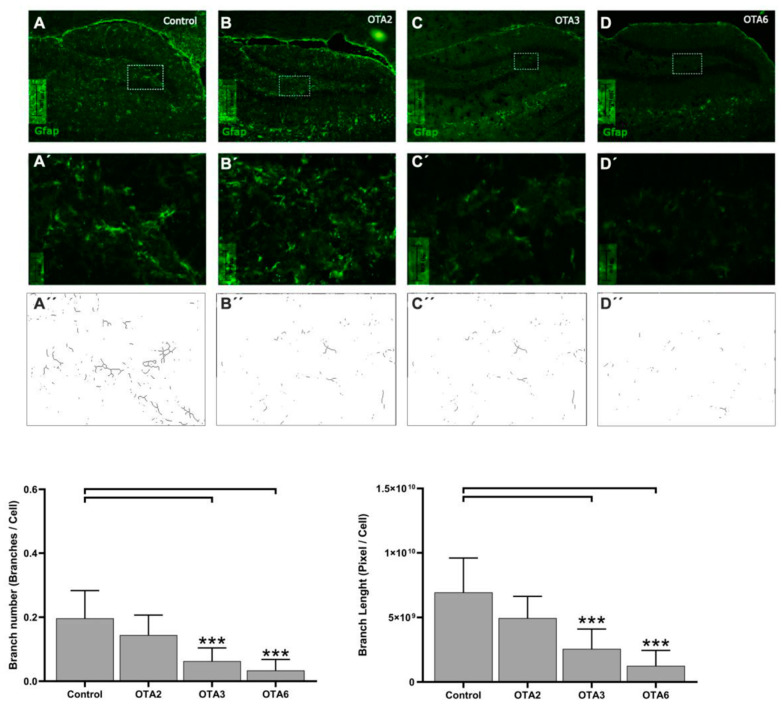
Photomicrographs of GFAP-labeled cells in the control, OTA2, OTA3, and OTA6 groups (which received two, three, or six ochratoxin A injections, respectively) at a 20× magnification Compared to the control: *** = *p* < 0.001. An example of the branch quantification and the branches per cell and branch lengths for each photo are also shown. (Control n = 4, OTA2 n = 4, OTA3 n = 4, OTA6 n = 5).

**Figure 5 toxins-14-00624-f005:**
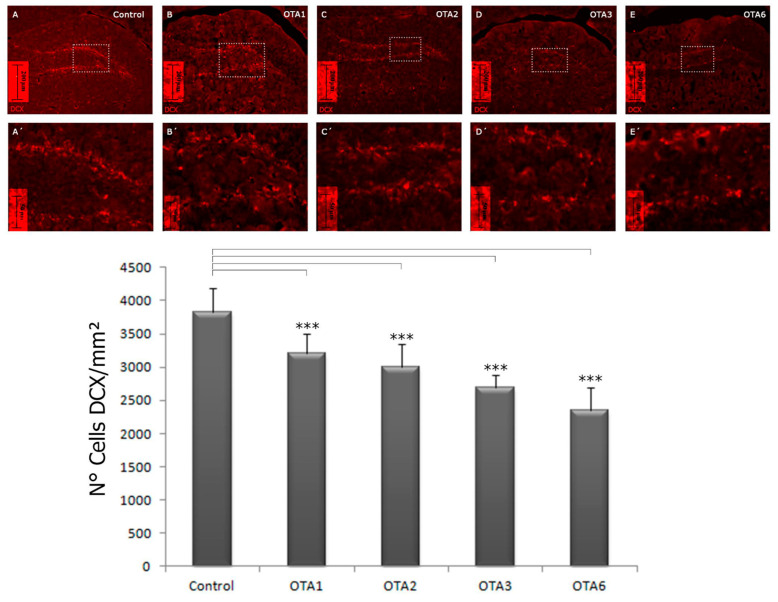
Photomicrographs of DCX-labeled cells in the dentate gyrus of the control, OTA1, OTA2, OTA3, and OTA6 groups (which received one to six injections with ochratoxin A, respectively) at a magnification of 20×. Compared to the control, the number of DCX-positive cells decreased as the number of ochratoxin A treatments increased: *** = *p* < 0.001. Games–Howell statistical analysis. (Control n = 4, OTA1 n = 4, OTA2 n = 4, OTA3 n = 4, OTA6 n = 5).

**Figure 6 toxins-14-00624-f006:**
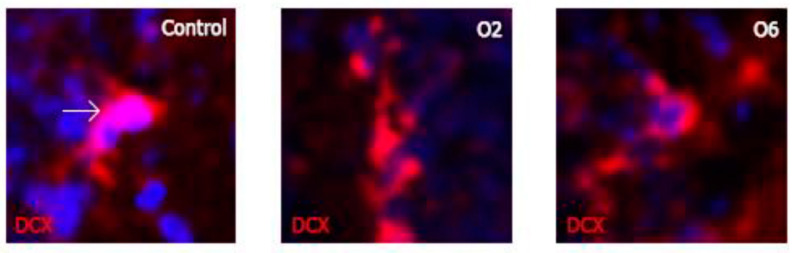
Photomicrographs of individual DCX-labeled young neurons demonstrating small morphological changes as the doses of ochratoxin A increased. The sizes of the cells appear to have decreased compared to the control group, although these changes do not appear to be as prominent as with the GFAP-labeled cells.

**Figure 7 toxins-14-00624-f007:**
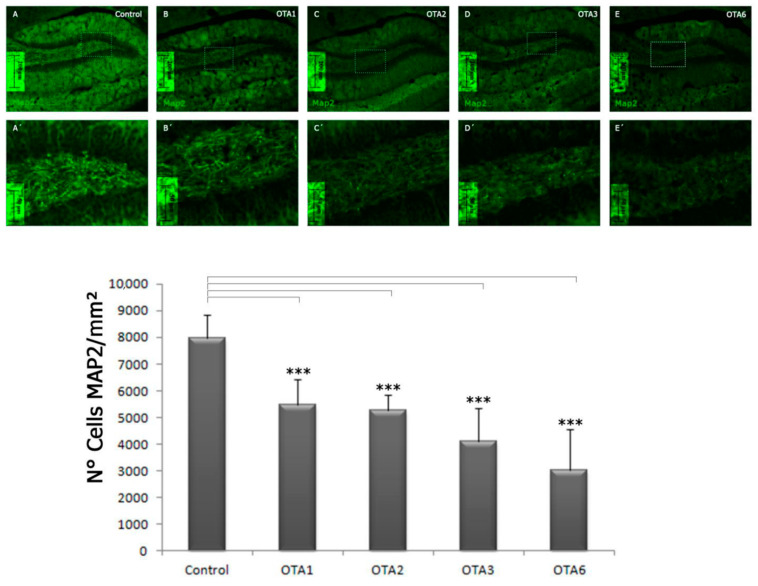
Photomicrographs of MAP2-labeled cells in the dentate gyrus of the control, O1, O2, O3, and O6 groups taken at a 20× magnification. Compared to the control, the number of MAP2-positive cells decreased as the number of ochratoxin A treatments increased. *** = *p* < 0.001). Statistical analysis was one-way ANOVA with a post-hoc LSD test. (Control n = 4, OTA1 n = 4, OTA2 n = 4, OTA3 n = 4, OTA6 n = 5).

**Figure 8 toxins-14-00624-f008:**
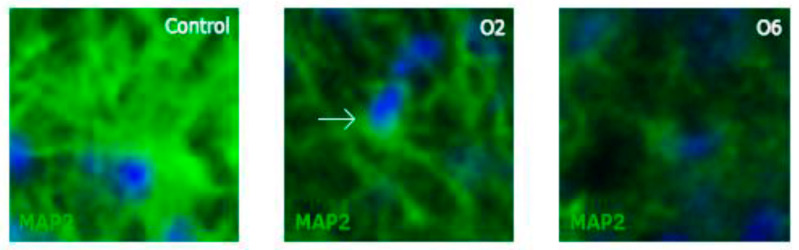
Photomicrographs of individual MAP2-labeled mature neurons and DAPI-labeled nuclei demonstrating their changing morphology as the number of doses of ochratoxin A increased. Whereas many dendrites and cells were present in the control, no comparable structures were present in the O6 group (6 doses of ochratoxin A). Very few dendrites could be found, and the cell body fluorescence was strongly decreased in the O6 group. The images were taken at a 40× magnification and digitally magnified to 400×. The arrow indicates a Map2/DAPI co-labeled cell.

**Table 1 toxins-14-00624-t001:** Number of DCX and DAPI co-labeled cells per square millimeter in the controls and OTA1, OTA2, OTA3, and OTA6 groups, which received one to six doses of ochratoxin-A, respectively. Statistical analysis was one-way ANOVA with a post-hoc LSD test. (Control n = 4, OTA1 n = 4, OTA2 n = 4, OTA3 n = 4, OTA6 n = 5).

	Control	OTA1	OTA2	OTA3	OTA6
*N* = cells/mm^2^	4	4	4	4	5
	3825 ± 182	3217 ± 1445	2999 ± 175	2686 ± 99	2343 ± 154
Significance compared to the control	-	<0.001	<0.001	<0.001	<0.001

**Table 2 toxins-14-00624-t002:** Numbers of MAP2 and DAPI co-labeled cells per square millimeter in the controls and OTA1, OTA2, OTA3, and OTA6 groups, which received one to six doses of ochratoxin-A, respectively. Statistical analysis was one-way ANOVA with a post-hoc Games−Howell test. (Control n = 4, OTA1 n = 4, OTA2 n = 4, OTA3 n = 4, OTA6 n = 5).

	Control	OTA1	OTA2	OTA3	OTA6
*N* = cells/mm^2^	4	4	4	4	5
	7990 ± 426	5466 ± 488	5247 ± 297	4095 ± 617	3006 ± 694
Significance compared to the control	-	<0.001	<0.001	<0.001	<0.001

**Table 3 toxins-14-00624-t003:** Primary and secondary antibodies used for fluorescence analysis and counting.

Product	Antibody	Dilution	Specificity	Wavelength	Reference	Company
H-1200	4′,6-diamidino-2-fenylindool (DAPI)	-	Binds with chromatin- DNA/RNA	461 nm (blue)	Chazotte 2011	Vector Labs, UK
Z0334	Glial fibrillary acidic protein (GFAP)	1:500	Astroglial lineage	-	Eng et al., 2000	Dako, Denmark
AB18723	Doublecortin (DCX)	1:300	Neuronal precursor cells and immature neurons	-	Gleave et al.	Abcam, UK
AB5622	Microtubule-associated protein 2 (MAP2)	1:200	Mature neurons	-	Lyck et al.	Millipore, CA, USA
A11008	Alexa Fluor™ 488 goat anti-rabbit	1:200	Secondary antibody	495–519 (green)	Borg et al.	Invitrogen, Spain
A1102	Alexa Fluor™ 594-conjugated donkey anti-rabbit	1:200	Secondary antibody	590–617 (red)	Purkartova et al.	Invitrogen, Spain

The combinations used were: GFAP—Alexa Fluor™ 488 (green), DCX—Alexa Fluor™ 594 (red), and MAP2—Alexa Fluor™ 488 (green). All the slides were mounted with DAPI (blue) in order to identify and count the nuclei.

## Data Availability

The data presented in this study are available on request from the corresponding author.

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
