# Peer review of "The Neurotoxic Effect of Ochratoxin-A on the Hippocampal Neurogenic Niche of Adult Mouse Brain"

_toxins, 2022, doi:10.3390/toxins14090624_

Round 1

Reviewer 1 Report

The paper's topic was the neurotoxic effect of ochratoxin-A on the hippocampal neurogenic niche of the adult mouse brain. The neurotoxic effect of OTA is known, but this paper adds some novel information about the toxic effects.

The paper contains a lot of editorial mistakes; for instance, there is many "Error! Reference source not found..." in the manuscript; therefore, it was not easy to find that the statements refer to which table of the figure.

The Discussion part is mostly hypothetical but acceptable. However, more relevant reference is required to support the explanation of the present study's data, even though few previous papers were published about this topic.

The Materials and Methods require double-check and editorial modification.

L 19-20: Six repeated doses are not chronic.

L 22-23: The critical contribution is missing.

L 29: The half-life of OTA is 800 hours in the food chain, but it is unclear which compartment.

L 41: Please, refer here that ochratoxin alpha is a nontoxic metabolite.

L 61-62: Please, modify as low as reasonably possible.

L 74: The dimension is not correct; it should modify to 3.5 mg/kg bw.

L 86: The number of samples should be added to each group separately because n=21 means the total number of animals in the study, including vehicle-treated control.

L 91-92: What means maintenance doses?

L 95-96: "all the groups significantly differed from each other (p < 0.01)" In Figure 2, significant differences are shown only as compared to the vehicle-treated control. Still, there are no differences shown between the groups.

L 100-101: Table 1 is a not necessary repetition of Figure 2 with numerical data. Please, keep only one, the figure or the table.

L 105-107: A description of the statistical evaluation should add to the figure legend.

L 120: The number of samples should add to each group separately because n=21 means the total number of animals in the study, including vehicle-treated control

L 119-120: Figure 4: OTA 1 group is missing; therefore, the explanation in the next paragraph cannot evaluate correctly due to lack of data.

L 120-121: The number of samples should add to each group separately because n=21 means the total number of animals in the study, including vehicle-treated control

L 135-137: A description of the statistical evaluation should add to the table's title.

L 142143: The number of samples in each group is missing.

L 149: Games-Howell – what does it mean? However, it explains later; therefore not necessary to mention it here.

L 153: The number of this table is false because it is Table 3.

L 164-166: The statistical evaluation description should be added to the table's title.

L 170. The number of this figure is false because it is Figure 6.

L 171: The number of samples should be added to each group separately because n=27 means the total number of animals in the study, including vehicle-treated control. Otherwise, n=27 is possibly false because only 21 animals were treated.

L 225: What means more chronic OTA doses?

L 228-231: This explanation is not easy to follow. A more detailed explanation requires.

L 232: The half-life of OTA in the body cannot be 800 hours. It is a misunderstanding or false data.

L 248-249: A comparison of the present study's data to hypoxia-reperfusion injury is interesting but hypothetical because hypoxia-reperfusion induces oxidative stress, but no experimental evidence suggests that oxidative stress occurs after OTA exposure in the brain.

L 287: What happened with the remaining five animals? Because most of the figures showed that the number of animals was 21, but some others were 26. Please, double-check the number of animals in each figure and table, including those where data of the OTA1 group were missing.

L 287: Please add the age and sex of the animals.

L 288: How can divide 26 animals if they are kept in cages and five in each?

L 292: Please, add the approval number.

L 297: How can you treat 17 animals if those were kept in cages and five in one cage? What was the cause that the number of animals in the experimental groups was not the same?

L 302: How many animals are treated as vehicle-treated control? Controls were treated once, but the OTA-treated groups were in the three-days interval. When were the vehicle controls exterminated?

L 312: Please, add a complete reference to the Paxinos G atlas.

L 314: Which type of Triton was used?

L 325: "boiled" – please, add the temperature.

L 329-331: Please, add the references as in other parts of the manuscript with numbers and add all the references to the reference list.

Author Response

We thanks the reviewer all suggestions and changes.

Here we provide a point- by -point response to the reviewer´s comments:

The paper's topic was the neurotoxic effect of ochratoxin-A on the hippocampal neurogenic niche of the adult mouse brain. The neurotoxic effect of OTA is known, but this paper adds some novel information about the toxic effects.

The paper contains a lot of editorial mistakes; for instance, there is many "Error! Reference source not found..." in the manuscript; therefore, it was not easy to find that the statements refer to which table of the figure

There were some problems with the exportation to mdpi format, we have solved this problem.

The Discussion part is mostly hypothetical but acceptable. However, more relevant reference is required to support the explanation of the present study's data, even though few previous papers were published about this topic.

It has been modified as the reviewer suggests

The Materials and Methods require double-check and editorial modification.

L 19-20: Six repeated doses are not chronic.

It has been modified as the reviewer suggests

L 22-23: The critical contribution is missing.

It has been added as reviewer suggests

L 29: The half-life of OTA is 800 hours in the food chain, but it is unclear which compartment.

As the reviewer suggested, the sentence is not clear enough therefore the paragraph has been modified and we added a new reference "OTA is rapidly absorbed and distributed but slowly eliminated and excreted leading to potential accumulation in the body, which is due mainly to binding to plasma proteins and a low rate of metabolism. Plasma half-life range from several days in rodents and pigs to several weeks in nonhuman primates and humans." (Schrenk D, et al,. 2020).

L 41: Please, refer here that ochratoxin alpha is a nontoxic metabolite.

As the reviewer suggest the text has been modified and we added new references

L 61-62: Please, modify as low as reasonably possible.

The text of the manuscript has been modified as the reviewer suggests.

L 74: The dimension is not correct; it should modify to 3.5 mg/kg bw.

The text of the manuscript has been modified as the reviewer suggests.

L 86: The number of samples should be added to each group separately because n=21 means the total number of animals in the study, including vehicle-treated control.

The text of the manuscript has been modified as the reviewer suggests.

L 91-92: What means maintenance doses?

As the reviewer suggests, we have changed the term maintenance by cumulative doses

L 95-96: "all the groups significantly differed from each other (p < 0.01)" In Figure 2, significant differences are shown only as compared to the vehicle-treated control. Still, there are no differences shown between the groups. 

The text of the manuscript has been modified as the reviewer suggests.

L 100-101: Table 1 is a not necessary repetition of Figure 2 with numerical data. Please, keep only one, the figure or the table.

As the reviewer suggests we could keep just figure 2 however we would prefer to maintain table 1 in the manuscript

L 105-107: A description of the statistical evaluation should add to the figure legend.

It has been done as the viewer suggests

L 120: The number of samples should add to each group separately because n=21 means the total number of animals in the study, including vehicle-treated control.

It has been done as the reviewer suggests

L 119-120: Figure 4: OTA 1 group is missing; therefore, the explanation in the next paragraph cannot evaluate correctly due to lack of data.

We do not show the data obtained in O1 since did not see significant changes in OTA1 vs OTA2.

L 120-121: The number of samples should add to each group separately because n=21 means the total number of animals in the study, including vehicle-treated control

It has been done as the reviewer suggests

L 135-137: A description of the statistical evaluation should add to the table's title.

It has been done as the reviewer suggests

L 142143: The number of samples in each group is missing.

It has been done as the reviewer suggests

L 149: Games-Howell – what does it mean? However, it explains later; therefore not necessary to mention it here.

We change the text in order to make it clearer. But as the reviewer said it is explained in detail later in the Material and method section. 

L 153: The number of this table is false because it is Table 3.

It has been modified as the reviewer suggests.

L 164-166: The statistical evaluation description should be added to the table's title.

Has been modified as the reviewer suggests.

L 170. The number of this figure is false because it is Figure 6.

It has been modified as the reviewer suggests

L 171: The number of samples should be added to each group separately because n=27 means the total number of animals in the study, including vehicle-treated control. Otherwise, n=27 is possibly false because only 21 animals were treated.

It has been modified as the reviewer suggests.

L 225: What means more chronic OTA doses?

It has been modified as the reviewer suggests.

L 228-231: This explanation is not easy to follow. A more detailed explanation requires.

In order to avoid misunderstanding, we have modified the text.

L 232: The half-life of OTA in the body cannot be 800 hours. It is a misunderstanding or false data.

In order to avoid misunderstanding, we have modified the text.

L 248-249: A comparison of the present study's data to hypoxia-reperfusion injury is interesting but hypothetical because hypoxia-reperfusion induces oxidative stress, but no experimental evidence suggests that oxidative stress occurs after OTA exposure in the brain.

The reviewer is completely right, however, we have highlighted these results due to the enormous similarity in the effect after cell deterioration. However, there are some authors who highlight the role of OTA in the generation of oxidative stress in neural cells in vitro. (Bhat PV et al., 2016)

L 287: What happened with the remaining five animals? Because most of the figures showed that the number of animals was 21, but some others were 26. Please, double-check the number of animals in each figure and table, including those where data of the OTA1 group were missing.

The reviewer is right. It is a mistake since the total number is 21 (C=4, OTA1= 4, OTA2=4, OTA 3= 4, OTA 6 =5), we have modified this data in all the manuscript.

L 287: Please add the age and sex of the animals.

It has been done as the reviewer suggests.

L 288: How can divide 26 animals if they are kept in cages and five in each?

It has been corrected as the reviewer suggests since we used 21 animals.

L 292: Please, add the approval number.

It has been done as the reviewer suggests.

L 297: How can you treat 17 animals if those were kept in cages and five in one cage? What was the cause that the number of animals in the experimental groups was not the same?

The reviewer is right. It is a mistake since the total number is 21 (Control=4, OTA1= 4, OTA2=4, OTA3= 4, OTA 6 =5). we have modified this data in all the manuscript The number of animals was advised by the ethics committee, a number of 5 animals was assigned to the group most sensitive to OTA, which was the OTA6 group.

L 302: How many animals are treated as vehicle-treated control? Controls were treated once, but the OTA-treated groups were in the three-days interval. When were the vehicle controls exterminated?We change the text in order to make it clearer. But as the reviewer said it is explained in detail later in the Material and method section. 

L 312: Please, add a complete reference to the Paxinos G atlas.

It has been modified as the reviewer suggests .

L 314: Which type of Triton was used?

It has been done as the reviewer suggests.

L 325: "boiled" – please, add the temperature.

It has been done as the reviewer suggests.

L 329-331: Please, add the references as in other parts of the manuscript with numbers and add all the references to the reference list.

It has been done as the reviewer suggests.

Reviewer 2 Report

The authors described the effect of ochratoxin A on the neurogenic niche of the hippocampus of the adult mouse brain. Unfortunately, the data is not very clear and it is very difficult to understand the results because the figures and tables are added in a disordered way. Furthermore, the values of P in different figures need to be revised because it is from the figures that the significance values of OTA at dose 1 and 2 do not appear graphically in agreement with the values of P.

Author Response

We thanks the reviewer for comments and suggestions

Here we provide a point-by -point response to the reviewer´s comments

The authors described the effect of ochratoxin A on the neurogenic niche of the hippocampus of the adult mouse brain. Unfortunately, the data is not very clear and it is very difficult to understand the results because the figures and tables are added in a disordered way. Furthermore, the values of P in different figures need to be revised because it is from the figures that the significance values of OTA at dose 1 and 2 do not appear graphically in agreement with the values of P.

The reviewer is absolutely right, the manuscript has been extensively proofread. Tables, figures and all references have been revised and text has been modified. Results have been described in greater detail, as well as the footnotes of figures and tables have been improved in order to make these results more understandable.

We thanks the reviewer and now we hope that manuscript will be suitable to be accepted

Reviewer 3 Report

The authors described the OTA neurotoxicity in adult mouse brain.

The document is interesting but more information is needed. In particular:

line 60: please provide the addition of the updated EU food recommendation

line 80: add the figure and the correct table

line 98: verify that the P is correct, from the digit appears incorrect to OTA 1 and OTA2 respect to control.

line 125: check the figure

Line 166-168: please check this sentence, because the data presented here are not concise and precise

Line 188: please add more reference (doi: 10.3390/antiox10010125; doi: 10.1002/jcp.26753.)

Line 194: motivate the dose of 3.5 mg / kg.

Line 220-222: please, rewrite in more correct English

Line 303-304: specify the sacrifice and add the reference for the choice of dose

Line 36: add some reference (doi: 10.3390 / antiox10081239.)

Author Response

We thanks the reviewer for suggestions and comments

Here we address point by point response to the reviewer´s comments

The authors described the OTA neurotoxicity in adult mouse brain.

The document is interesting but more information is needed. In particular:

line 60: please provide the addition of the updated EU food recommendation.

It has been done as the reviewer suggests.

line 80: add the figure and the correct table

 It has been done as the reviewer suggests.

line 98: verify that the P is correct, from the digit appears incorrect to OTA 1 and OTA2 respect to control.

It has been done as the reviewer suggests.

line 125: check the figure

It has been done as the reviewer suggests.

Line 166-168: please check this sentence, because the data presented here are not concise and precise

It has been done as the reviewer suggests.

Line 188: please add more reference (doi: 10.3390/antiox10010125; doi: 10.1002/jcp.26753.)

It has been done as the reviewer suggests.

Line 194: motivate the dose of 3.5 mg / kg.

“This dose was selected because it is in the range of doses used by other studies, in this way the obtained results could be compared easily [27,40].”

Line 220-222: please, rewrite in more correct English.

It has been done as the reviewer suggests.

Line 303-304: specify the sacrifice and add the reference for the choice of dose

It has been done as the reviewer suggests.

Line 36: add some reference (doi: 10.3390 / antiox10081239.)

It has been done as the reviewer suggests.

Thanks again for suggestions and comments and now we hope the manuscript would be suitable to be accepted

Best and thank you agan

Reviewer 4 Report

The manuscript entitled “The neurotoxic effect of ochratoxin-A on the hippocampal neurogenic niche of adult mouse brain” is not suitable for publication in TOXINS in the current format, because of the following reason.

1.      The study lacks behind the nobility.

2.      The results are not well explained.

3.      Figure legends are not well explained.

4.      The experiments performed in the study are very basic and preliminary. 

Round 2

Reviewer 1 Report

The revised version of the manuscript improved. The Authors corrected most  of the critical items as I suggested. However, the Discussion part remains hypothetical and some relevant references are missing.

In Figure 4 Group 1 is missing and there is no explanation about its cause. 

The manuscript requires accurate editorial corrections.

Author Response

Comments and Suggestions for Authors

The revised version of the manuscript improved. The Authors corrected most of the critical items as I suggested. However, the Discussion part remains hypothetical and some relevant references are missing.

As the reviewer suggests,  Introduction and Discussion have been modified and new citations have been incorporated

In Figure 4 Group 1 is missing and there is no explanation about its cause. 

We thank the reviewer for the observation. In our experiments represented in the figure 4, we have not seen any difference between Group1 and Group 2, for this reason, we decided not to add this data to the figure. In addition, other papers from our group have seen that there are not any significant differences between these groups [1,2].

  1. Erceg, S.; Mateo, E.M.; Zipancic, I.; Jiménez, F.J.R.; Aragó, M.A.P.; Jiménez, M.; Soria, J.M.; Garcia-Esparza, M.Á. Assessment of Toxic Effects of Ochratoxin A in Human Embryonic Stem Cells. Toxins (Basel). 2019, 11, doi:10.3390/TOXINS11040217.
  2. Paradells, S.; Rocamonde, B.; Llinares, C.; Herranz-Pérez, V.; Jimenez, M.; Garcia-Verdugo, J.M.; Zipancic, I.; Soria, J.M.; Garcia-Esparza, M.A. Neurotoxic effects of ochratoxin A on the subventricular zone of adult mouse brain. J. Appl. Toxicol. 2015, 35, 737–751, doi:10.1002/JAT.3061.

The manuscript requires accurate editorial corrections.

As the reviewer suggests English has been revised by a Scientific editing enterprise" (Scientific Editing- EFL)- Maria Ledran- Carrer 227, 20 pta3 Paterna Valencia Spain

Reviewer 2 Report

ready to publish, regards

Author Response

Comments and Suggestions for Authors

Ready to publish, regards.

We thank the suggestions of the reviewer, however, some changes regarding English have been done.

Thanks again

Reviewer 3 Report

please, check english. After which, the article is ready for publication

Author Response

Comments and Suggestions for Authors

Please, check your English. After which, the article is ready for publication

As the reviewer suggests English has been revised by a Scientific editing enterprise" (Scientific Editing- EFL)- Maria Ledran- Carrer 227, 20 pta. 3 Paterna Valencia Spain

However we will change or modify any suggestion from the editorial office.

Thanks again

Reviewer 4 Report

I think the manuscript entitled “The neurotoxic effect of ochratoxin-A on the hippocampal neurogenic niche of adult mouse brain” has been widely improved than before. However, it needs further revision before publication.

1.      I think the introduction needs more information on the environmental bioavailability of OTA.

2.      What is the relevance of the concentration chosen in the study with the concentration of toxins present in the environment?

3.      What do you mean by “OTA, as shown in Error!” in L293.

4.      I think authors should provide the H&E staining data to indicate the changes in the tissue morphology upon OTA treatment.

5.      I strongly recommend modifying the introduction section of the manuscript. The below mentioned papers are suitable for citation:

Dey et al. 2022. Crit Rev Food Sci Nutr. 1-22.

Kőszegi and Poór. 2016. Toxins (Basel). 111.

Bui-Klimke and Wu. 2015. Crit Rev Food Sci Nutr. 1860–1869.

Author Response

Comments and Suggestions for Authors

I think the manuscript entitled “The neurotoxic effect of ochratoxin-A on the hippocampal neurogenic niche of adult mouse brain” has been widely improved than before. However, it needs further revision before publication.

  1. I think the introduction needs more information on the environmental bioavailability of OTA.

Introduction have been modified and new citations have been incorporated

  1. What is the relevance of the concentration chosen in the study with the concentration of toxins present in the environment?

It has been described in material and method: This dose was selected because it is in the range of doses used by other studies, in this way the obtained results could be compared easily. (citations are in the reference list). More concretely, Sava et al. (2006a,2006b) demonstrated that the administration of OTA at a single dose (3.5 mg kg1), approximately 10% of the reported LD50, resulted in widespread oxidative stress in all six brain subdivisions considered macroscopically (cerebellum, hippocampus, caudate putamen, pons medulla, midbrain and cerebral cortex).

  1. What do you mean by “OTA, as shown in Error!” in L293.

The text of the manuscript have been modified

  1. I think authors should provide the H&E staining data to indicate the changes in the tissue morphology upon OTA treatment.

Experiments with hematoxylin and eosin were performed, however, no significant structural changes were observed in general, this is the reason because they have not been reported. in this manuscript. However, we agree with the reviewer, and new experiments using hematoxylin and eosin and even ultrastructural studies using electron microscopy would be very useful and will be developed in future studies. We modified the discussion regarding this fact.

In discussion we have added: "In this study, experiments with hematoxylin and eosin were performed, and no significant macro-structural changes were observed, however new studies should be performed using electron microscopy in order to study the utra structure of the hippocampus after OTA exposition"

  1. I strongly recommend modifying the introduction section of the manuscript. The below mentioned papers are suitable for citation:

Dey et al. 2022. Crit Rev Food Sci Nutr. 1-22.

Kőszegi and Poór. 2016. Toxins (Basel). 111.

Bui-Klimke and Wu. 2015. Crit Rev Food Sci Nutr. 1860–1869.

We thank the suggestions of the reviewer; we have modified introduction and we have included these suitable citations